

# Microstructure representation of snow in coupled snowpack and microwave emission models

Melody Sandells[1], Richard Essery[2], Nick Rutter[3], Leanne Wake[3], Leena Leppänen[4], and
Juha Lemmetyinen[5]

[1]CORES Science and Engineering Limited, Burnopfield, UK
[2]Edinburgh University, Edinburgh, UK
[3]Northumbria University, Newcastle-upon-Tyne, UK
[4]Finnish Meteorological Institute, Arctic Research Centre, Sodankylä, Finland
[5]Finnish Meteorological Institute, Helsinki, Finland

*Correspondence to:* Melody Sandells (melody.sandells@coresscience.co.uk)

**Abstract.** This is the first study to encompass a wide range of coupled snow evolution and microwave emission models in a common modelling framework in order to generalise the link between snowpack microstructure predicted by the snow evolution models and microstructure required to reproduce observations of brightness temperature as simulated by snow emission models. Brightness temperatures at 18.7 and 36.5GHz were simulated by 1323 ensemble members, formed from 63 Jules Investigation Model snowpack simulations, three microstructure evolution functions and seven microwave emission model configurations. Two years of meteorological data from the Sodankylä Arctic Research Centre, Finland were used to drive the model over the 2011-2012 and 2012-2013 winter periods. Comparisons between simulated snow grain diameters and field measurements with an IceCube instrument showed that the evolution functions from SNTHERM simulated snow grain diameters that were too large (mean error 0.12 to 0.16mm), whereas MOSES and SNICAR microstructure evolution functions simulated grain diameters that were too small (mean error -0.16 to -0.24mm for MOSES, and -0.14 to -0.18mm for SNICAR). No model (HUT, MEMLS or DMRT-ML) provided a consistently good fit across all frequencies and polarizations. The smallest absolute values of mean bias in brightness temperature over a season for a particular frequency and polarization ranged from 0.9 to 7.2K.

Optimal scaling factors for the snow microstructure were presented to compare compatibility between snowpack model microstructure and emission model microstructure. Scale factors ranged between 0.3 for the SNTHERM-Empirical MEMLS model combination (2011-2012), and 5.0 or greater when considering non-sticky particles in DMRT-ML in conjunction with MOSES or SNICAR microstructure (2012-2013). Differences in scale factors between microstructure models were generally greater than the differences between microwave emission models, suggesting that more accurate simulations in coupled snowpack-microwave model systems will be achieved primarily through improvements in the snowpack microstructure representation, followed by improvements in the emission models. Other snowpack parameterisations in the snowpack model, mainly densification, led to a mean brightness temperature difference of 11K when the JIM ensemble was applied to the MOSES microstructure and empirical MEMLS emission model for the 2011-2012 season. Consistency between snowpack



microstructure and microwave emission models, and the choice of snowpack densification algorithms should be considered in the design of snow mass retrieval systems and microwave data assimilation systems.

# 1 Introduction

Global observations of the snow cover extent from optical and microwave satellite observations combined with in-situ data have shown a reduction in the spring snow cover (Brown et al., 2010; Brown and Robinson, 2011). Observed decline in snow cover extent during 2008-2011 exceeded that predicted by climate models (Derksen and Brown, 2012). Observations also indicate that duration of snow cover is also reducing, but cannot determine whether mass or volume of snow has changed.

Microwave and coarser-scale gravity satellite sensors offer the only way to measure snow mass or depth on a global scale, but microwave algorithms such as those developed by Chang et al. (1987); Kelly (2009) can result in large errors because of the because of the high sensitivity of applied forward models to parameterization of the snow microstructure (Davenport et al., 2012). In particular, the assumption of a fixed snow scatterer radius in the Chang et al. (1987) algorithm does not reflect the naturally changing snowpack structure. Errors in snow mass products derived from these algorithms mean that the products are difficult to use for evaluation of snow mass in climate models (Clifford, 2010) and unsuitable for assimilation into land surface models for streamflow forecasts (Andreadis and Lettenmaier, 2006). Development of the assimilation-based technique in GlobSnow allows changes in the snow microstructure to be taken into account through inversion of ground-based observations of snow depth and coinciding microwave brightness temperatures (Takala et al., 2011). Although more accurate than other global products, some errors remain, and the GlobSnow accuracy relies on the proximity and representativity of the ground stations (Hancock et al., 2013). In addition, the intermediate retrieval of the snow 'grain size' in GlobSnow is a parameter that also incorporates other land surface features, so is not a true representation of the snow effective diameter (Lemmetyinen et al., 2015).

Snowpack evolution models offer a way to estimate temporal changes in snow microstructural parameters and stratigraphy (e.g. Lehning et al., 2002; Brun et al., 1992). Intercomparison studies have shown large differences between snow evolution models driven by the same forcing data (Rutter et al., 2009). Given that the mass inputs were the same for the 33 snow models considered in the SNOWMIP2 study of Rutter et al. (2009), it is differences in the internal snow physics and model structure (layering assumptions) that result in the wide range of simulated depth and snow mass. Temperature, temperature gradient and density drive changes in the snow microstructure (e.g. Flanner and Zender, 2006), so it is likely that different snow physics assumptions in a coupled snowpack and emission model result in different thermal structures, microstructure parameterisations and ultimately different microwave extinction behaviour.

Theoretical differences between specific electromagnetic models have been examined in Löwe and Picard (2015); Pan et al. (2015), and other intercomparisons carried out Tedesco and Kim (2006). These studies are useful for interpreting differences in electromagnetic model outputs for a snapshot profile of the snowpack properties. Given the dependence of microwave scattering on snow microstructure, a satellite retrieval system needs some quantification of microstructure. Snowpack evolution modelling offers a means to quantify the metamorphic changes in snow microstructure. Indeed, snowpack evolution models have been



coupled with microwave emission models to demonstrate the potential of this approach for snow remote sensing applications (Langlois et al., 2012; Andreadis and Lettenmaier, 2012; Brucker et al., 2011; Picard et al., 2009). These studies all examined the accuracy of a single snowpack model coupled with a single microwave emission model.

The purpose of this study is to inform future design of retrieval and assimilation systems where snowpack evolution models
may be used to provide microstructural parameters for microwave emission models, by examining how particular snowpack and emission model choices lead to a variation in simulated brightness temperatures throughout the winter period, and evaluate how the simulated values compare to observations. The Jules Investigation Model (Essery et al., 2013) has been coupled with three widely used microwave emission models: The Dense Media Radiative Transfer Model - Multilayer (Picard et al., 2013), the Microwave Emission Model of Multi-Layer Snow (Wiesmann and Mätzler, 1999) and the Helsinki University of
Technology Multilayer model (Lemmetyinen et al., 2010; Pulliainen et al., 1999). Snowpack microstructure metamorphism is represented here by three different options with differing complexity for grain diameter evolution (or equivalently the Specific Surface Area). These models are the grain growth models of SNTHERM (Jordan, 1991), SNICAR (Flanner and Zender, 2006) and MOSES (Essery et al., 2001). This allowed quantification of the seasonal variation in uncertainty in brightness temperature simulations from 1,323 coupled snowpack-emission model systems, as evaluated against ground based observations of
brightness temperature.

The study approach, model descriptions and field measurements are given in section 2. Comparisons between simulations, and between simulations and observations are presented in section 3, and the implications for future approaches to the remote sensing of snow mass are discussed in section 4.

## 2 Models and methods

This study builds on the work of Essery et al. (2013), who incorporated many published snow model parameterisations (excluding microstructural representation) within a single model framework, the Jules Investigation Model (JIM), which is described in section 2.1. For this study, JIM was coupled with three microstructure evolution functions, described in section 2.2 and three distinct snow emission models, detailed in section 2.3. Steps necessary to form the model ensemble, including assumptions about the representation of the soil are given in section 2.4. A description of the field site, driving and evaluation data for the
simulations in this paper are presented in section 2.5.

### 2.1 Snow model parameterisation

Essery et al. (2013) developed the Jules Investigation Model (JIM), a system of 1701 snowpack evolution models to provide a systematic method and common framework to examine how the range of snowpack parameterisations used in land surface models impacts the simulation of snow parameters. Based on this work, a more computationally efficient version, the Factorial
Snowpack Model, has been developed (Essery, 2015), which allows for 32 model configurations. JIM is based on an Eulerian grid scheme (fixed layer structure), which requires mass redistribution between layers with precipitation events. An alternative approach is a Lagrangian grid scheme: a deforming layer structure that retains much of the same snow material throughout the





season (e.g. Jordan, 1991; Brun et al., 1992; Lehning et al., 2002). For this paper, a subset of the original JIM members were selected as these were expected to influence the parameters important for microwave modelling. The subset includes variation in the representation of compaction, the density of newly-deposited snow, thermal conductivity and liquid water flow (snow hydrology). Table 1 summarizes the different approaches taken. Note that a variable fresh snow density (options 0 and 1) cannot be used if the snowpack has fixed density (compaction option 2), so there are only 63 model configurations in the model subset rather than 81. For all other snowpack parameterisations, option '1' from Essery et al. (2013) were used for albedo, surface exchange and snow fraction representations to form the JIM subset.

## 2.2 Microstructure evolution

JIM subset outputs were used to drive three microstructure models of differing complexity. SNTHERM (SNT) (Jordan, 1991) growth of snow grain diameter $d$ is based on the rate of vapour transport through the snow (and therefore temperature gradient), which leads to the microstructure evolution function of dry snow in SNT as:

$$\frac{\partial d}{\partial t} = \frac{g1}{d} D_{eos} \left(\frac{1000}{P_a}\right) \left(\frac{T_s}{T_m}\right)^6 C_{kT_s} \left|\frac{\partial T_s}{\partial z}\right| \tag{1}$$

where $g1$ and $D_{eos}$ are empirical constants, $P_a$ is the atmospheric pressure, $C_{kT_s}$ is the variation of saturation vapour pressure with snow temperature $T_s$, $T_m = 273.15K$ and $\frac{\partial T_s}{\partial z}$ is the temperature gradient. Grain growth under wet conditions is more rapid, with empirical constant $g2$ and is dependent on the liquid fractional volume, $f_l$ by:

$$\frac{\partial d}{\partial t} = \frac{g2}{d} (f_l + 0.05) \qquad\qquad\qquad\qquad f_l < 0.09 \tag{2}$$

$$\frac{\partial d}{\partial t} = \frac{g2}{d} (0.14) \qquad\qquad\qquad\qquad f_l \geq 0.09 \tag{3}$$

SNICAR (SNI) microstructure evolution is a computationally efficient approximation to a model based on physics, and uses a look-up table for empirical parameters $\tau$ and $\kappa$, as described in Flanner and Zender (2006). These parameters are dependent on the snow density, temperature, and temperature gradient. The equation of microstructure evolution in SNI is based on snow specific surface area (SSA):

$$SSA(t) = SSA_0 \left(\frac{\tau}{t+\tau}\right)^{1/\kappa} \tag{4}$$

SSA per unit mass of ice (m$^2$ kg$^{-1}$) can then be converted to grain diameter with $D = 6/(\rho_i\, SSA)$ (Mätzler, 2002; Montpetit et al., 2012).

A third microstructure model, MOSES (MOS), parameterizes snow evolution as a function of grain radius $r$ and snow age:

$$r(t+\Delta t) = \left[r(t)^2 + \frac{G_r}{\pi} \Delta t\right]^{1/2} - [r(t) - r_0]\frac{S_f \Delta t}{d_0} \tag{5}$$

where $G_r$ is an empirical temperature-dependent grain area growth rate, $S_f$ is the snowfall rate in time interval $\Delta t$ and $d_0$ is a constant representing the mass of fresh snow required to reset the snow albedo to its maximum value.





Other microstructure parameterisations are available, namely the Crocus (Vionnet et al., 2012) and SNOWPACK (Lehning et al., 2002) microstructure evolution functions. It is not currently possible to couple these with the JIM model due to the Eulerian grid structure of JIM. Mass transfer between layers allows numerical averaging of concepts such as grain diameter and SSA, but not shape-dependent concepts such as dendricity and sphericity. Therefore the Crocus and SNOWPACK functions have not been included in this study.

## 2.3 Microwave emission models

The microwave models chosen for this application span a range of physical complexity in their representation of the snow. The Helsinki University of Technology (HUT) model (Lemmetyinen et al., 2010) is a semi-empirical model based on strong forward scattering assumptions, the Microwave Emission Model of Multi-Layer Snow (MEMLS) model (Wiesmann and Mätzler, 1999) is of intermediate complexity and contains the Improved Born Approximation (Mätzler, 1998), and the Dense Media Radiative Transfer Multilayer (DMRT-ML) model (Picard et al., 2013) is the most physically complex, being based on quasi-crystalline approximation with coherent potential (QCA-CP). Many other microwave emission models have been developed, such as Mie scattering approach of Boyarskii and Tikhonov (2000); Chang et al. (1976); Eom et al. (1983); strong fluctuation theory (Stogryn, 1986; Song and Zhang, 2007), distorted Born approximation (Tsang et al., 2000), the quasi-crystalline approximation (Grody, 2008), other QCA-CP models Rosenfeld and Grody (2000); Jin (1997) or the Numerical Method of Maxwell's equations in 3D (Xu et al., 2012). These references are not exhaustive but give an illustration of the range of models available. Here, we restrict the comparison to widely available multilayer models that span a range of complexity and whose computational efficiency is such that entire seasons can be simulated rapidly.

Of the models chosen, all are multiple layer and broadly require the same information i.e. they use layered information on snow temperature, density and layer thickness as input, but differ in their representation of the microstructure. They are all based on radiative transfer theory, which is governed by the following general equation:

$$\mu \frac{\partial \mathbf{T}_B(\theta_s, \phi_s, z)}{\partial z} = \kappa_a T(z) + \frac{1}{4\pi} \int_{4\pi} \Psi(\theta_s, \phi_s; \theta_i, \phi_i) \cdot \mathbf{T}_B(\theta_i, \phi_i, z) \, d\Omega - \kappa_e \cdot \mathbf{T}_B(\theta_s, \phi_s, z) \qquad (6)$$

where $\theta$ and $\phi$ are the incidence and azimuth angles, $\mu = \cos \theta$, $\mathbf{T}_B$ is the brightness temperature vector, which we will assume here to consist of horizontally and vertically polarized brightness temperature only, $\kappa_a$ is the absorption coefficient, $\kappa_e$ is the extinction coefficient, which is a sum of the absorption coefficient and the scattering coefficient $\kappa_s$. The models differ in which two coefficients determine the third. In HUT, the derived coefficient is $\kappa_s$, whereas $\kappa_e$ is derived in MEMLS and $\kappa_a$ in DMRT-ML. Other differences between models include the representation of the phase function (single-stream model with separate up- and downwelling components in HUT, 6-stream in MEMLS and multiple streams in DMRT-ML), specification of the absorption coefficient and the numerical techniques applied to solve the radiative transfer equation (Lemmetyinen et al., 2010; Wiesmann and Mätzler, 1999; Picard et al., 2013; Mätzler and Wiesmann, 1999; Pan et al., 2015). Differences between models are not restated here, but options chosen within each model leading to different model versions are stated in the following subsections.





### 2.3.1 DMRT-ML

DMRT-ML is based on a sticky hard spheres representation of the microstructure so that the scattering coefficient given by the quasi-crystalline approximation with coherent potential is given as:

$$\kappa_s = \frac{2}{9} k_0^4 a^3 f \left| \frac{\epsilon_s - \epsilon_b}{1 + \frac{\epsilon_s - \epsilon_b}{3E_{eff}(1-f)}} \right|^2 \frac{(1-f)^4}{(1+2f-tf(1-f))^2} \tag{7}$$

where $k_0 = 2\pi/\lambda$ is the wave number, a is the radius of the spheres, $f$ is the fractional volume of scatterers, $\epsilon_s$ is the permittivity of the scatterers, $\epsilon_b$ is the permittivity of the background and $E_{eff}$ is the effective permittivity of the medium. $t$ is related to the stickiness factor $\tau$ governing the potential of particles to coalesce. For non-sticky particles t=0 but for sticky particles, it is given by the largest solution to

$$\frac{f}{12} t^2 - \left( \tau + \frac{f}{1-f} \right) t + \frac{1+f/2}{(1-f)^2} = 0 \tag{8}$$

Whilst Löwe and Picard (2015) have shown that it may be possible to determine stickiness from micro-CT measurements of the snow, an appropriate value of stickiness is not known for the field observations used in this paper. For this model ensemble, two DMRT-ML varieties have been chosen to capture the range of brightness temperatures simulated: "DMRT non-sticky" ($\tau = 10^6$) and "DMRT sticky" ($\tau = 0.1$).

### 2.3.2 MEMLS

Within MEMLS there are a suite of options for the calculation of the scattering coefficient. Two of the options within MEMLS were selected for this study to cover both empirical and theoretical approaches: "MEMLS empirical" and "MEMLS IBA". The empirical version of MEMLS used gives the scattering coefficient as:

$$\kappa_s = (9.2p_{ec} - 1.23\rho + 0.54)^{2.5} (\nu/50)^{2.5} \tag{9}$$

where the correlation length $p_{ec}$ is in mm, density $\rho$ is in g cm$^{-3}$ and frequency $\nu$ is in GHz. This is suitable for correlation lengths $0.05 < p_{ec} < 0.3$ mm, and density $0.1 < \rho < 0.4$ g cm$^{-3}$.

MEMLS IBA uses the Improved Born Approximation theory given in Mätzler (1998); Mätzler and Wiesmann (1999), where the scattering coefficient is given by the integral of the phase function for an averaged polarization angle $\chi$

$$\kappa_s = \frac{1}{4\pi} \int_{4\pi} f(1-f)(\epsilon_s - 1)^2 K^2 I k_0^4 \sin^2 \chi \, d\Omega \tag{10}$$

Averaging of the polarization angle means that it is only dependent on the incidence angle $\theta$, not the azimuth, so that:

$$\sin^2 \chi = 1 - \frac{1}{2} \sin^2 \theta \tag{11}$$

One further assumption applied to distinguish this MEMLS IBA configuration is that oblate grains are used rather than small spherical scatterers or thin spherical shells. This assumption governs the representation of the mean squared field ratio, $K^2$ as





detailed in Mätzler and Wiesmann (1999). The microstructure length information is contained in $I$:

$$I = \frac{2p_{ec}}{\left(1 + 4\epsilon_{eff}k_0^2 \sin^2(\theta/2)p_{ec}^2\right)^2} \tag{12}$$

It should be noted that the choice of oblate grains also affects the effective permittivity in $I$.

### 2.3.3 HUT

HUT has three options for the extinction coefficient. These are nominally suited to different grain size ranges, with some overlap between them. All three versions (termed "HUT H87", "HUT R04", "HUT K10") have been included in this version of the model ensemble. HUT H87 is based on the work of Hallikainen et al. (1987):

$$\kappa_e = 0.0018\nu^{2.8}d_0^{1.9} \tag{13}$$

This is nominally appropriate for frequency range $\nu$=18-60GHz and $d_0 < 1.6$ mm.

The extinction coefficient in HUT H04, with a validity range of $1.3 < d_0 < 4$ mm was derived by Roy et al. (2004):

$$\kappa_e = 2\nu^{0.8}d_0^{1.2} \tag{14}$$

Kontu and Pulliainen (2010) gave the extinction coefficient for maritime snow, used here in the HUT K10 simulations as:

$$\kappa_e = 0.08\nu^{1.75}d_0^{1.8} \tag{15}$$

Scaling of the grain diameter by the relationship recommended in Kontu and Pulliainen (2010) has not been applied here as it was developed for snow microstructure observations rather than simulated snowpack microstructure.

### 2.4 Model framework

Interfacing of the various model inputs and outputs was enabled through the development of the ensemble framework, via a combination of shell script and Octave/Matlab code. Internal parallelization of the MATLAB code of HUT-MEMLS means that a season-long simulation of one HUT-MEMLS combination with one grain scaling factor takes 9 minutes over eight cores. For the DMRT-ML FORTRAN code, external bash shell parallelization reduces execution time from 16 hours to circa 2 hours for one grain scale factor and two parameterisations of stickiness. Over 29 million individual brightness temperatures were simulated for this study.

For the purposes of this study, the effective sphere size in JIM, DMRT-ML and HUT are assumed to be identical i.e. $d_{hut} = 2 \times r_{dmrt} = d_{jim}$. This may not be a good assumption as the empirical extinction coefficient model used in HUT was based on observations of the maximum grain extent rather than effective diameter, which was almost impossible to measure at the time of the original work. The exponential correlation length in MEMLS (in mm) is calculated from the theoretical relationship to the effective grain diameter from JIM (in microns) as Montpetit et al. (2013); Mätzler (2002):

$$p_{ec} = \frac{2}{3}\left(1 - \frac{\rho_s}{\rho_i}\right)\frac{d_{jim}}{1000} \tag{16}$$



Figure 1 illustrates the flow of information in the model ensemble. Meteorological data are used to drive the 189 flavours of JIM. The outputs from JIM are then reformatted for each of the three electromagnetic models. Table 2 gives a summary of the main differences in inputs between models. The electromagnetic model inputs are then used to drive the two DMRT-ML model versions ($\tau$=0.1, $\tau = 10^6$), the two MEMLS model versions (empirical, IBA with oblate grains) and the three HUT versions (3 different extinction coefficient models). Meteorological and field data used to drive and evaluate the ensemble are described in the following section.

## 2.5 Data

Model runs for this study were performed for the Intensive Observation Area (IOA) of the Finnish Meteorological Institute Arctic Research Centre (FMI-ARC). The site provides a wealth of forcing and evaluation data, including automated soil, snow and meteorological observations, ground-based microwave radiometry, and a programme of manual snow profile observations. Air temperature, solar radiation and precipitation observations from this site for the two seasons of simulations are shown in Figure 2. Metadata and details on the meteorological instruments are given in Essery et al. (2016). Dual polarization microwave radiometers, including at frequencies of 18.7 and 36.5 GHz are situated on a 4m tower pointing inwards on the edge of a large clearing surrounded by a mainly pine forest. Further details about the IOA site are given in Lemmetyinen et al. (2016). Details on the manual snow profile observation programme are given by Leppänen et al. (2016).

Simulations were carried out for the winters of 2011-2012 and 2012-2013 as there where 49 approximately bi-weekly snow pit observations over these two years available for snowpack model evaluation. Snow samples from 31 of these pits were extracted and used to measure profiles of the snow specific surface area (SSA) with the IceCube instrument (Zuanon, 2013). A bulk grain diameter was calculated for the analysis from the SWE-weighted mean SSA, excluding layers without observations. For these two seasons, the real component of the soil permittivity measurements were available at 100MHz, measured at three locations in the observation area with Delta-T Devices ML2x sensors, installed horizontally at a depth of approximately 2 cm beneath the organic surface layer. Mean measurements from the stable winter period (1st December to 31st March) were chosen as representative for the entire season, which resulted in values of soil permittivity of 4.4 in 2011-2012 and 4.6 in 2012-2014. For the JIM simulations in this paper, a scaling factor of 1.11 was applied to the 2011-2012 precipitation data, and a scaling factor of 1.06 was applied to the 2012-2013 data to match the measured snow accumulation on the ground better. These factors differ slightly from the values used in the 7-year consolidated dataset of Essery et al. (2016).

## 2.6 Simulation methodology

Choices in the snowpack evolution parameterisations made here lead to 189 unique JIM snowpack models. Modelled snowpack profiles of layer thickness, temperature, density and grain diameter were output daily at noon for this study. These were then applied to the seven microwave emission model combinations, resulting in 1,323 sets of brightness temperature simulations per day.

In order to illustrate and analyze the effects of assumptions regarding snowpack evolution and microwave scattering on simulated brightness temperatures over the course of winter season, the remainder of the paper will:





1. Present the range of brightness temperatures expected for any generic combination of snowpack and emission model.

2. Apply a range of scaling factors ($0.1 <= \Phi <= 5.0$) to simulated JIM snowpack diameters (equation) and calculate the degree of misfit between simulated and observed brightness temperatures using the following cost function (CF):

$$CF = \sum^{ndays} \sum^{\nu} \sum^{pol} \left( \frac{TB_{sim} - TB_{obs}}{2} \right)^2 \tag{17}$$

The cost function term is summed over the two polarizations (H and V pol) for the two frequencies (18.7 and 36.5 GHz) over the number of days (ndays) where observations and simulations are both available. Due to the observation schedule at the Sodankylä site, the noon 'observations' for comparison with the simulations were determined as the mean of the 10am and 2pm observations. If observations were missing from either or both of these times, the brightness temperature for that day was excluded from the cost function calculation. Optimal $\Phi$ were found from the minimisation of the cost function CF.

3. Isolate the effect of snowpack parameterizations on simulated brightness temperature by presenting simulation results grouped by parameterizations of densification, liquid water flow, initial snow density and thermal conductivity. This will determine which factors govern the spread in brightness temperature and are therefore important for the design of snow retrieval assimilation systems.

## 3   Results

Snow depth and snow water equivalent (SWE) simulated by the Jules Investigation Model is shown in Figure 3. There is a small difference between the automatic measurements and the manual field observations attributable to the spatial variability of the snow and difference in measurement location. Ultrasonic snow depth measurements were on average 12mm deeper than the snow pit observations in 2011-2012 but were 29mm shallower than snow pit measurements in 2012-2013. SWE measured automatically by the gamma ray sensor had a mean value of 3.6mm SWE greater than the pit observations in 2011-2012 but 5.9mm less in 2012-2013.

Although the precipitation inputs were scaled due to known sensor undercatch problems, in 2011-2012 the SWE was underestimated until the end of January, then overestimated until the melt period. Compared with snow pit observations, the SWE bias prior to 1st February was -14.2 mm. Between 1st February and 31st March, the SWE bias was 19.12mm. From 1st April until the end of the season, the bias was -24.51 mm water equivalent. In 2012-2013, simulated SWE was overestimated for most of the season, with a mean bias of 13.73mm compared with the snow pit observation. Simulated SWE is relatively insensitive to the snow parameterisation in the accumulation period, but 3 distinct model groups emerge in the melt period, which are due to the 3 different representations of the liquid water flow. The snowpack model parameterisations have a greater impact on the snow depth, which is to be expected as this is directly affected by the representation of densification and initial snow density.

Figure 4 demonstrates the impact of the snow model parameterisations on snow grain diameter growth, as simulated with the MOS, SNI and SNT microstructure evolution functions. Each microstructure model results in a spread of bulk grain diameter



due to the 63 snowpack parameterisations, but in general the difference between microstructure models is greater than the difference due to snowpack parameterisations. The simulation range is greatest at the start and at the end of the season, when the snowpacks can be subject to the largest temperature gradients, or liquid-water dependent growth. In both years of simulation, the mid season bulk grain diameter is smallest with MOS, and largest with SNT. MOS and SNIs are similar in

magnitude, but SNT bulk grain diameters were approximately twice as large on average, with a mean ratio over the season of 1.9-2.2, as shown in Table 3. SNT bulk grain diameter was up to 3.2 times larger than MOS bulk grain diameter. Visual estimation of the snow grain diameter gave values that were always larger than all of the simulations. Measured SSA-derived bulk grain diameters generally lay in between the SNT simulations and the simulations with SNI and MOS. The mean absolute error for these simulations are presented in Table 4. SNT had the lowest bias (0.12mm) in 2011-2012, whereas SNI had the

lowest bias (-0.14mm) in 2012-2013.

Simulation of mean and range of brightness temperature from the three emission models driven by all snowpack and microstructure model combinations is shown in Figure 5. Note that excessively low brightness temperatures on 01 November 2011 were excluded from this figure as the snowpack for some JIM members was extremely thin with an unphysically high snow density. In general, HUT with 3 representations of extinction coefficient showed the smallest range of brightness tem-

perature, whereas DMRT-ML (covering both sticky and non-sticky spherical particle assumptions) had a much greater range, which was nearly as large as MEMLS (empirical representation and Improved Born Approximation with oblate grains). This is demonstrated by the ratio between the seasonal mean ranges of brightness temperature presented in Table 5, where the ranges compared with HUT had a ratio of greater than 1. MEMLS had a larger range than DMRT-ML, although at 37GHz the difference was small. As illustrated in Figure 5, at 19GHz, the mean of DMRT-ML simulations were highest and the mean

of MEMLS simulations were generally lowest (with the exception of 19H in 2011-2012). At 37 GHz, horizontal polarization, HUT gives the highest mean brightness temperature in 2012-2013, whereas DMRT-ML gives the highest mean brightness temperature in 2011-2012 and at horizontal polarization (both years). MEMLS mean brightness temperatures are the lowest at 37GHz at both horizontal and vertical polarization in both years. All ranges exhibit a distinctive 'wedge' shape, where the ranges generally increase throughout the season until the collapse of the range in the melt period.

Compared with the brightness temperature observations, no model gives a consistently better performance across both frequencies and both polarizations. This is illustrated in Table 6, where mean bias and root mean squared error (RMSE) for each season has been presented for each frequency and polarization combination. The lowest bias was less than 8K in magnitude, whereas the lowest RMSE for each frequency / polarisation was less than 13K. For both years, DMRT-ML gave the lowest bias at 19H and MEMLS gave the lowest bias at 37GHz (V and H). At 19V DMRT-ML had the lowest bias in 2011-2012 whereas

HUT had the lowest bias in 2012-2013. Figure 5 shows that the observed brightness temperature is generally within the range simulated by each of the three microwave emission models, with the exception of 19H in 2011-2012 (MEMLS and HUT) and 37H in 2012-2013 (HUT). End of season brightness temperature observations are not replicated in the simulations as the liquid water content of the snowpack model is currently decoupled from the electromagnetic snow model, so the simulations only represent dry snow brightness temperature.



Table 7 indicates scaling factors that would need to be applied to the grain diameter in order to allow a particular microstructure evolution function to minimize the cost function given in equation 17 i.e. the best agreement with observed brightness temperature for all four frequency and polarization combinations. A scale factor of 1 suggests a perfect fit between snowpack microstructure and microwave microstructure. A scale factor of less than one indicates a snowpack grain diameter overestimate,

whereas a scale factor of greater than one is an underestimate. For SNT microstructure, a scale factor of less than one was required in 2011-2012 for all emission models with the exception of the non-sticky application of DMRT-ML. This indicates that the SNT microstructure resulted in grain diameters larger than that required by the emission models for that year. In 2012-2013 SNT microstructure required slight scaling to increase the grain diameter for HUT, but downscaling for sticky hard spheres in DMRT-ML and for MEMLS. With the exception of the application to empirical MEMLS in 2011-2012, the SNI and MOS

grain diameters were too small and required scaling upwards. A cost function minimum was achieved for empirical MEMLS driven by MOS microstructure with no scaling whatsoever in 2011-2012. In contrast, large scaling factors were required with the assumption of non-sticky particles in DMRT-ML. For MOS and SNI in 2012-2013 the cost function was minimised by a scale factor of 5 (or greater), but even the grain diameters simulated with SNT were too small. The pattern is consistent between years, with the greatest interannual difference in scale factor for HUT and for DMRT-ML non-sticky microstructure

from MOS.

Once the microstructure differences have been isolated through application of the optimal scale factor, except for 37H in 2012-2013, the lowest RMSE in brightness temperature is reduced, as shown in Table 8. DMRT-ML bias and RMSE improved with the application of the optimal scale factor, with the exception of the small increase in 19V bias in 2011-2012. For MEMLS, improvements in bias and RMSE at one frequency / polarisation were at the expense of another frequency / polarisation in both

years. This was also the case for HUT in 2012-2013, whereas in 2011-2012 the bias and RMSE decreased at all frequencies and polarisations apart from a marginal (<0.04K) increase in RMSE at 19V.

Differences in brightness temperature also exist in the simulations due to the snowpack parameterisation (i.e. 63 JIM combinations). Empirical MEMLS with MOS microstructure in the 2011-2012 season was chosen as a test case to illustrate the effects of snowpack parameterisation on the brightness temperature, because of the equivalence of snowpack and emission

model microstructure (no scaling required). This subset of 63 simulations for 37H brightness temperature in 2011-2012 is shown in Figure 6. There is a seasonal dependence in the range, with model divergence from mid-January onwards. February 1st and May 1st were chosen for cluster analysis to determine which parameterisations caused the split in simulations, as shown in Figure 7.

Clear groupings of simulations in Figure 7, upper left, indicates that the snowpack densification parameterisation has a

distinguishable effect on the simulation of brightness temperature. A physical representation of densification (parameterization = "0") gave the lowest brightness temperatures on the 1st February, but the highest by 1st May. In contrast, where no compaction is simulated i.e. snow density is constant throughout the season (parameterization = "2"), the opposite is true. An empirical representation of densification (parameterization = "1") results in brightness temperatures generally between those of the physical, and of no densification. Thermal conductivity has no effect on the simulation of brightness temperature, whereas

subtle differences are attributable to the fresh snow density value, and to the representation of snow hydrology. There is no

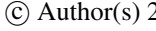



discernible difference between fresh snow density parameterisation schemes "0" and "1", whereas "2" gives a different set of brightess temperatures. Snow hydrology has very little effect in the early season, but can lead to differences in the melt period. Overall, the snowpack parameterisations with MOSES microstructure and empirical MEMLS lead to a mean difference of 11K and maximum difference of 33K in brightness temperature simulation at 37H for the 2011-2012 season.

**4   Discussion**

The biggest difference to obtaining accurate simulations would be made by improving the microstructure evolution models within snowpack models because the optimal scale factors are generally larger between microstructure models than between emission models. SNTHERM grains tend to be too large for the emission models and generally require scaling down to smaller values. SNICAR grains are in the mid-range and require a small amount of scaling, generally upwards to larger grains. MOSES

grains are the smallest and generally require larger scale factors than SNICAR. These patterns are consistent, regardless of the electromagnetic radiative transfer model used. Differences between microstructure evolution models are so large because they were developed in models with different purposes. MOSES is a large scale land surface model, requiring snow grain size for albedo calculations. SNICAR was also developed with an albedo focus. SNTHERM, on the other hand, was developed to predict surface temperature and uses grain diameter in the simulation of liquid water flow as well as albedo. SNICAR and

MOSES grain sizes are closer to the SSA-derived grain diameter as a result. SNTHERM simulates a grain size that is closer in concept to the visual estimates of grain diameter than the other two models. The large spread when coupling snowpack evolution and microwave models, due to the differences in the modelling of snow microstructure is consistent with the wide range of studies that have investigated how to link snowpack observations of microstructure to the microstructure parameter required in electromagnetic models (e.g Kendra et al., 1998; Du et al., 2005; Tedesco et al., 2006; Liang et al., 2008; Durand

et al., 2008; Brucker et al., 2011; Xu et al., 2012; Montpetit et al., 2013; Roy et al., 2013; Rutter et al., 2014; Picard et al., 2014).

Nevertheless there are differences between microwave emission models for a particular microstructure evolution model, and even differences within the same family of emission models. 'Improvement' in the microstructure for a particular model combination may lead to less accurate simulations at some frequencies and polarisations, which highlights that there is more to

understand. DMRT-ML is the only emission model of the three that has a polarisation-dependent phase function, which could partially explain why the reductions in bias and RMSE were nearly universal for DMRT-ML with application of a scale factor, but were frequency / polarization dependent for MEMLS or HUT.

For DMRT-ML, consideration of the stickiness is imperative, with the choice of microstructure evolution model of secondary importance. Only two extremes were considered here: extremely cohesive, or individual particles. The true value is likely to lie

between these. Löwe and Picard (2015) have made progress in understanding how to retrieve stickiness from micro-CT data, but more needs to be done in order to relate stickiness to other snow properties that are measurable in the field.

For the HUT radiative transfer model, the optimum combinations of snowpack and microwave model are dependent on both models, and therefore the end application. The SNT microstructure is most closely matched to the microstructure of the



Hallikainen et al. (1987) extinction model. Both were developed with a similar concept of microstructure. With MOS or SNI microstructure, Roy et al. (2004) would be most appropriate. Kontu and Pulliainen (2010) is more broadly applicable as the scale factor always lies between R04 and H87, regardless of the microstructure model. Therefore K10 may be better choice if a range of microstructure models are considered in a data assimilation retrieval scheme but with only one observation operator.

In the case of MEMLS, there are some differences between the empirical model, and IBA, but the microstructure model really matters. IBA is a more appropriate model for the larger SNT grains and endorses the recommendation of Mätzler and Wiesmann (1999) for IBA in the simulation of larger grains. The microstructural concept of MOS matches the microstructure of the empirical very well, with no scaling required in 2011-2012, although SNI is equally appropriate in 2012-2013.

There is little variation between years for the DMRT-ML (sticky) and MEMLS models, and a consistent pattern for HUT.
Other studies have investigated the microstructural link between snowpack and microwave models. Wiesmann et al. (2000) found that the scale factor between exponential correlation length in MEMLS and grain diameter in SNTHERM for the Weiss-fluhjoch site in Davos, Switzerland, was 0.16. Applying equation 16 for snow of density 250 kg m$^{-3}$, the scale factor to relate the grain diameter of SNTHERM to the exponential length in MEMLS for the Sodankylä site would be 0.24 for IBA and 0.15 for empirical MEMLS for the 2011-2012 dataset. This is entirely consistent with the Wiesmann et al. (2000) study, in spite of
the different locations and snowpack conditions.

Wiesmann et al. (2000) also reported a relationship for Crocus simulations, as did Brucker et al. (2011). At this stage it is not possible to make comparisons of this work with those studies as the Crocus evolution model has not been included in this study, due to the difficulty of applying these models to the Eulerian frame snowpack model scheme used here. These two studies are, however, consistent with each other. Wiesmann et al. (2000) found a snow type dependent scale factor of 0.3-0.4
between MEMLS correlation length and Crocus grain diameter, whereas the range in Brucker et al. (2011) was 0.25-0.4 for snow density between 100 and 400 kg m$^{-3}$. The scaling factor between the SNOWPACK-derived correlation length and the correlation length of MEMLS was found to be 0.1 (Langlois et al., 2012) but again, a comparison with this work is not possible as the SNOWPACK grain evolution model has similar requirements to the Crocus microstructure model as they have a common origin.

When isolating the spread in brightness temperature due to snowpack parameterisations, this spread is largely due to the snowpack model representation of the densification process, with a variable impact throughout the season. After the microstructure model, snow compaction must be considered carefully in the design of a coupled snowpack-microwave model as it can account for differences in brightness temperature of the order of 30K. Liquid water flow representation in the snowpack model may become important in the melt period, particularly for a snowpack with mid-winter melt periods or if the snowpack model
is used to provide information on SWE during melt when microwave observations cannot. If fresh snow is assumed to have a constant density in a retrieval or assimilation system then that value will have an impact but is less important than compaction. Thermal conductivity has no discernable impact on the brightness temperature simulations so the choice of its representation is largely irrelevant for snow mass retrieval and assimilation systems.

Although the differences in scale factors between microstructure models are larger than the differences in scale factors
between microwave models, this does not negate the need for developments in the microwave models. Use of scale factors can





improve brightness temperature accuracy at some frequency and polarisations, but may decrease the accuracy at others. The necessity of scale factors indicate the need for a deeper understanding in the role of microstructure in the microwave models. Much of this is discussed from a theoretical perspective by Löwe and Picard (2015). With a sticky hard sphere model of the microstructure, even if the stickiness is known, Löwe and Picard (2015) showed that a scale factor to relate the measured optical

diameter to microwave diameter depends on the type of metamorphism the snow has been subjected to. Indeed, here, constant scale factors have been applied with no attempt to assess how these may change over the season. Nor do they account for the anisotropic nature of the snow, which adds to the complexity both in the modelling of the snowpack (Löwe et al., 2013) and in microwave scattering (Leinss et al., 2015). Some of the fundamental questions on how to relate snowpack and microwave microstructure may be addressed with a better microstructure descriptor of the snowpack rather than a single length scale,

and would benefit from easy interchangability between different microwave models, and different snowpack evolution models. Ultimately a consistent microstructural treatment will be needed in both snowpack evolution and microwave models.

## 5   Conclusions

Future snow mass and depth retrievals systems may rely on snowpack models to provide snow microstructural parameters. To improve accuracy in seasonal simulations of brightness temperature, the largest gains will be achieved by improving the

microstructural representation within snowpack models, followed by improvements in the emission models to use accurate microstructural information and reduce bias and RMSE at all frequencies and polarizations simultaneously. For the design of retrieval systems with current capabilities, particular model combinations may be more suitable than others, and careful consideration must be given to snow compaction processes. The future lies in a better and consistent treatment of snow microstructure in both snowpack and emission model developments.

*Acknowledgements.* This work was funded in part by the NERC National Centre for Earth Observation, and supported by the European Space Agency projects "Technical assistance for the deployment of an X- to Ku-band scatterometer during the NoSREx experiment" (ESA ESTEC contract 22671/09/NL/JA/ef) and "Microstructural origin of electromagnetic signatures in microwave remote sensing of snow" (ESA ESTEC contract 4000112698/14/NL/LvH). We thank the staff of FMI Arctic Research Centre in Sodankylä for performing the in situ measurements.



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



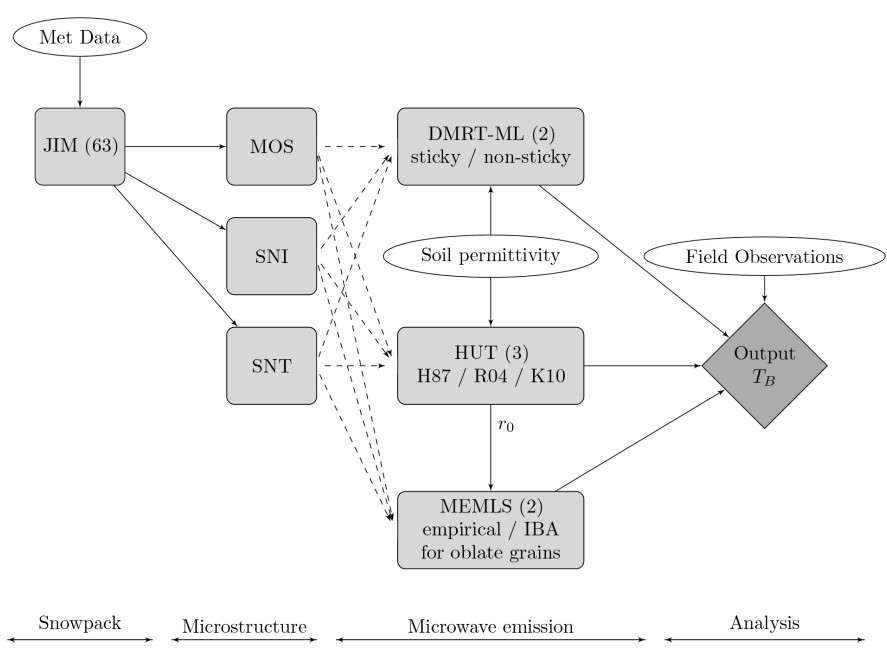

**Figure 1.** Flowchart showing flow of information from JIM snow evolution model outputs to outputs from the various microwave emission models.





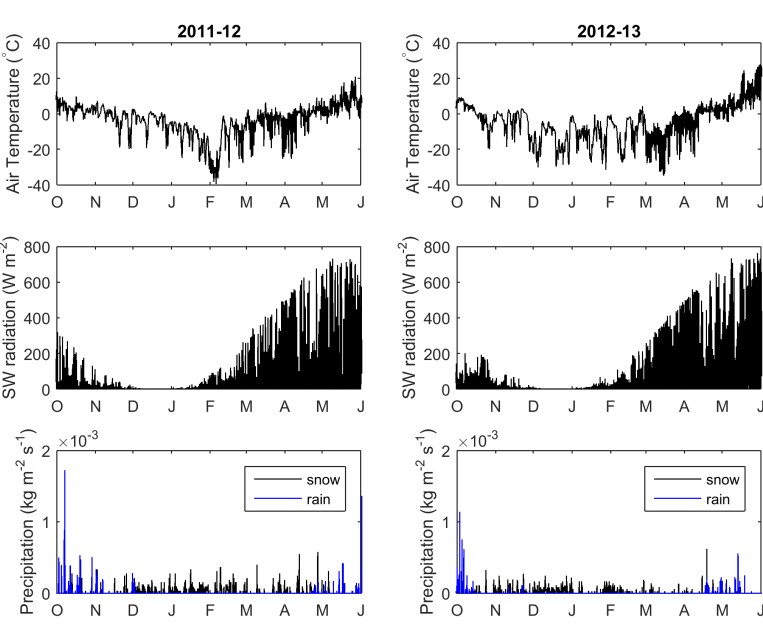

**Figure 2.** Air temperature, solar radiation and precipitation data measured at the Sodankylä site, used as inputs for the JIM simulations.





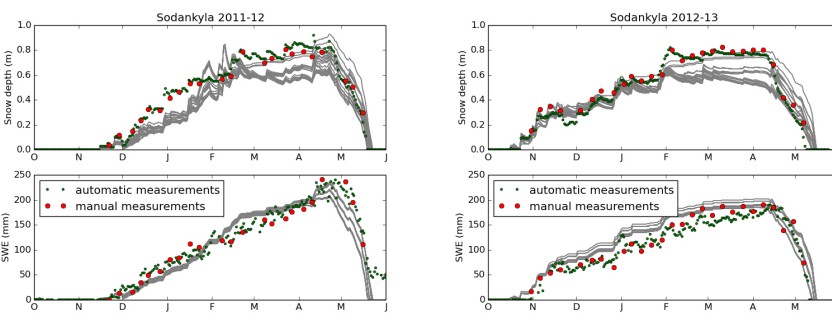

**Figure 3.** Snow depth and water equivalent simulated by the Jules Investigation Model subset used in this study. Grey lines indicate individual JIM subset member simulations.





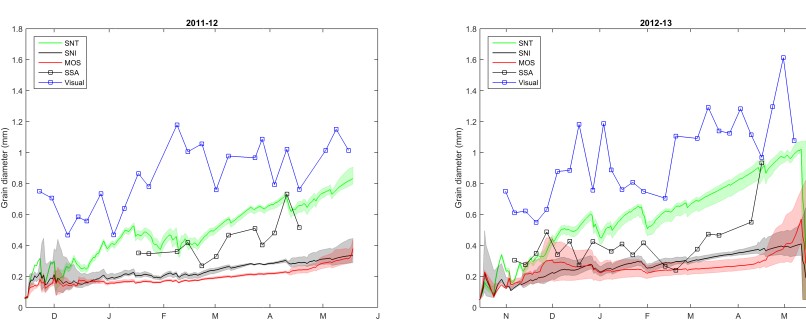

**Figure 4.** Bulk grain diameter evolution for the MOS, SNT and SNI microstructure evolution models and the spread in model results. Observations of bulk diameter were derived from macro-photography (Visual) and from SSA measurements from the IceCube instrument



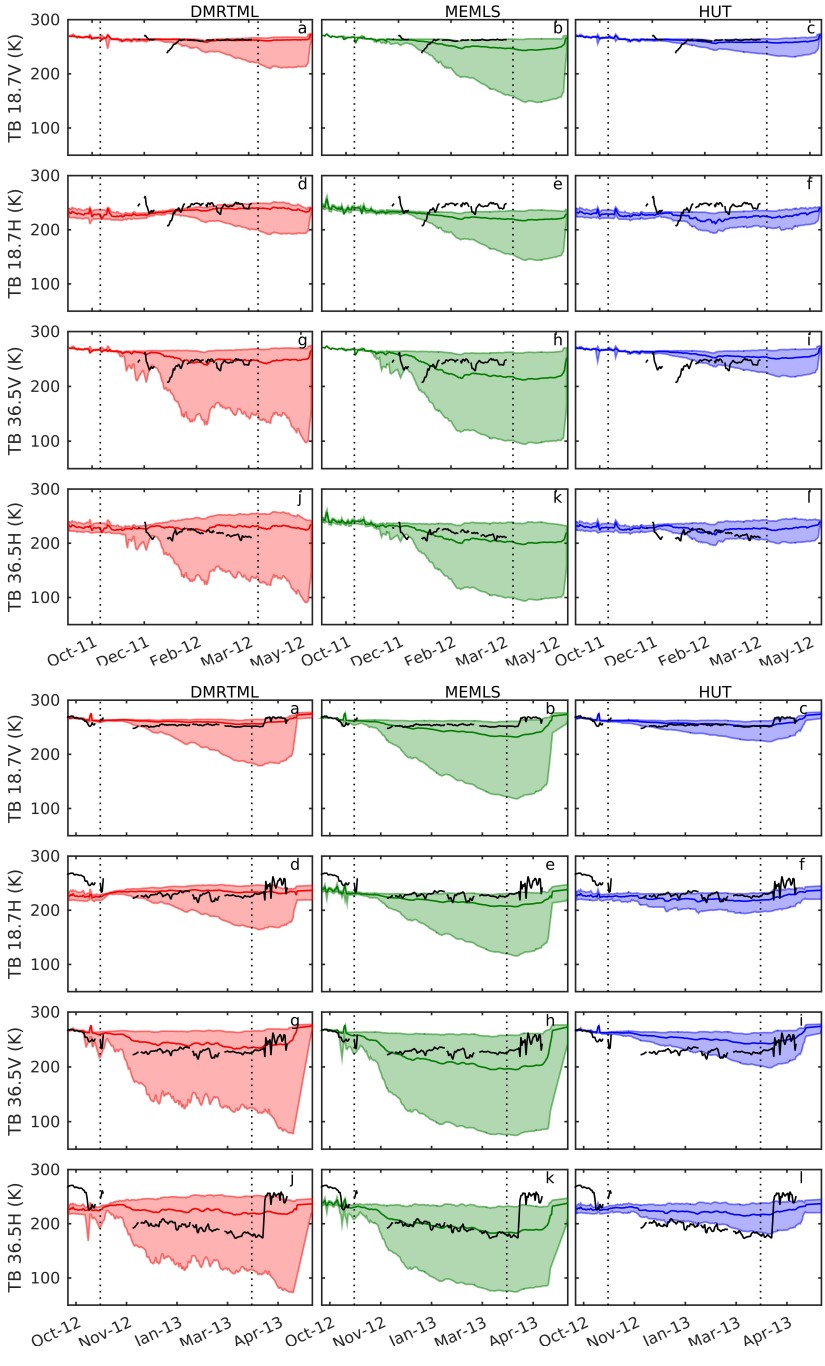

**Figure 5.** Range and mean of brightness temperature over the two winter seasons as simulated with the DMRT-ML, MEMLS and HUT models, driven by 63 JIM outputs and 3 microstructure evolution models. Black lines indicate the observed brightness temperatures. Vertical dashed lines enclose the period of analysis (1st November - 31st March).





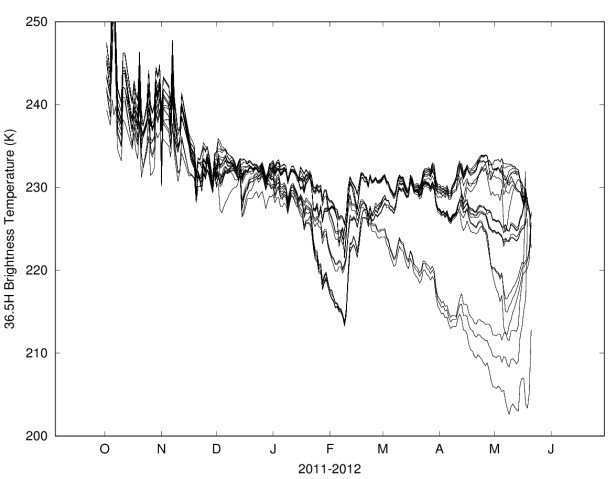

**Figure 6.** Variability in brightness temperature simulated with empirical MEMLS, driven by the MOSES microstructure model and 63 JIM snowpack outputs (no scaling of microstructure was required).





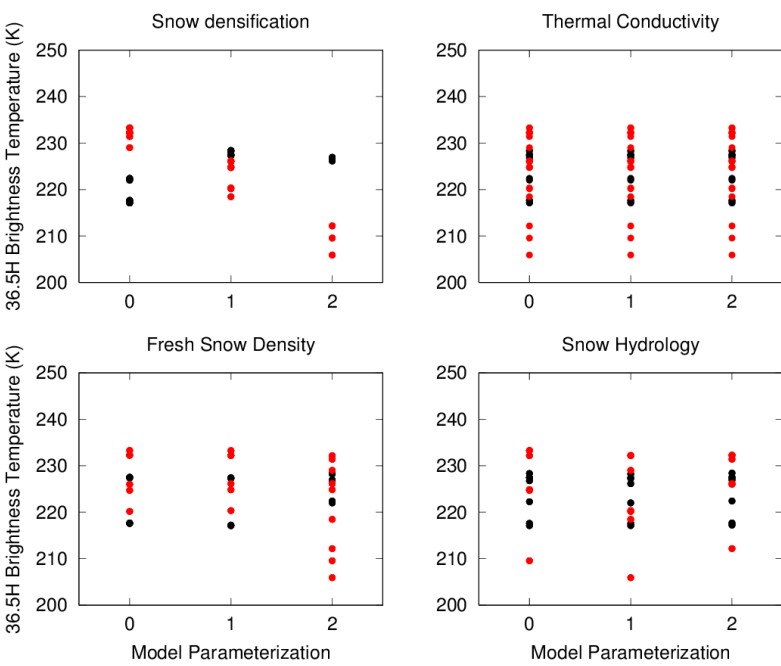

**Figure 7.** Cluster analysis of brightness temperature simulated by MOSES microstructure and empirical MEMLS for 1st February 2012 (black dots) and 1st May 2012 (red dots) according to model parameterization choices. Values 0, 1, 2 relate to parameterizations given in Essery et al. (2013) and as described in section 2.1.





**Table 1.** Equations for options governing the representation of processes in the JIM model subset

| Option | Description | Model Parameterization |
|---|---|---|
| Compaction: 0 | Physical | $\frac{1}{\rho_s}\frac{d\rho_s}{dt} = \frac{M_s g}{\eta} + c_1 \exp[-c_2(T_m - T_s) - c_3 \max(0, \rho_s - \rho_0)]$ |
| 1 | Empirical | $\rho_s(t + \delta t) = \rho_{max} + [\rho_s(t) - \rho_{max}]\exp(-\delta t/\tau_\rho)$ |
| 2 | Constant | $\rho_s = 250$ kg m$^{-3}$ |
| Fresh snow density: 0 | Empirical | $\rho_f = \max[a_f + b_f(T_a - T_m) + c_f U_a^{1/2}, \rho_{min}]$ |
| 1 | Empirical | $\rho_f = \rho_{min} + \max[d_f(T_a - T_m + e_f)^{3/2}, 0]$ |
| 2 | Constant | $\rho_f = 100$ kg m$^{-3}$ |
| Thermal conductivity: 0 | Empirical | $\lambda_s = \lambda_a + (a_\lambda \rho_s + b_\lambda \rho_s^2)(\lambda_i - \lambda_a)$ |
| 1 | Empirical | $\lambda_s = c_\lambda \left(\dfrac{\rho_s}{\rho_w}\right)^{n_\lambda}$ |
| 2 | Constant | $\lambda_s = 0.265$ W m$^{-1}$ K$^{-1}$ |
| Maximum liquid water: 0 | Empirical | $\dfrac{\gamma_{w,max}}{\rho_s} = r_{min} + (r_{max} - r_{min})\max\left(1 - \dfrac{\rho_s}{\rho_r}, 0\right)$ |
| 1 | Constant | $\gamma_{w,max} = \rho_w\left(1 - \dfrac{\gamma_i}{\rho_i}\right)S_{wi}$ |
| 2 | None | $\gamma_{w,max} = 0$ kg m$^{-3}$ |

JIM variables are: snow density $\rho_s$, overlying snow mass $M_s$, snow temperature $T_s$, air temperature $T_a$, wind speed $U_a$, snow effective thermal conductivity $\lambda_s$, partial density of liquid water $\gamma_w$, partial density of ice $\gamma_i$. Other symbols represent constants, given in Essery et al. (2013)





**Table 2.** Electromagnetic model inputs, as a function of JIM snowpack model outputs

|  | JIM output | DMRT-ML input | MEMLS input | HUT input |
|---|---|---|---|---|
| Temperature | $T_{jim}$ [K] | $T_{jim}$ [K] | $T_{jim}$ [K] | $T_{jim}$ - 273.15 [°C] |
| Density | $\rho_{jim}$ [kg m$^{-3}$] | $\rho_{jim}$ [kg m$^{-3}$] | $\rho_{jim}$ [kg m$^{-3}$] | $\frac{\rho_{jim}}{1000}$ [g cm$^{-3}$] |
| Layer size | $\Delta z_{jim}$ [m] | $\Delta z_{jim}$ [m] | $\frac{\Delta z_{jim}}{100}$ [cm] | $\rho_{jim}\Delta z_{jim}$ [mm$_{swe}$] |
| Microstructure | $d_{jim}$ [$\mu$m] | $d_{jim}$ [$\mu$m] | $\frac{2}{3}\left(1 - \frac{\rho_s}{\rho_i}\right)\frac{d_{jim}}{1000}$ [mm] | $\frac{d_{jim}}{1000}$ [mm] |
| Layer number | 1 = base | 1 = base | 1 = base | 1 = top |
| Soil permittivity | - | $\epsilon_{obs}$ | $r_{0,HUT}$ | $\epsilon_{obs}$ |

Note that for the purposes of the ensemble, DMRT-ML was adapted to allow the input of diameter rather than radius. HUT was adapted to ensure Fresnel reflectivity for a smooth soil surface, and to output the soil reflectivity $r_{0,HUT}$ at both polarizations for use in MEMLS simulations.





**Table 3.** Comparison between grain diameter ratio simulated by different microstructure models. Mean ratios and maximum grain diameter ratios for 2011-2012. Where the 2012-2013 values differ, these are given in parentheses.

|           | Mean      | Max       |
|-----------|-----------|-----------|
| SNI / MOS | 1.2 (1.1) | 1.4 (1.3) |
| SNT / MOS | 2.2       | 3.1 (3.2) |
| SNT / SNI | 1.9 (2.0) | 2.5       |



**Table 4.** Mean absolute error (mm) between bulk grain diameter simulated with the microstructure models compared with observations derived from SSA measurements with IceCube. Smallest bias for each year is shown in bold.

|     | 2011-2012 | 2012-2013 |
| --- | --- | --- |
| MOS | -0.24 | -0.16 |
| SNI | -0.18 | **-0.14** |
| SNT | **0.12** | 0.16 |





**Table 5.** Ratio of mean brightness temperature ranges simulated by two microwave emission models

| 2011-2012 | 19V | 19H | 37V | 37H | 2012-2013 | 19V | 19H | 37V | 37H |
|---|---|---|---|---|---|---|---|---|---|
| DMRTML / HUT | 1.6 | 1.2 | 3.8 | 3.1 | DMRTML / HUT | 2.0 | 1.6 | 3.3 | 3.0 |
| MEMLS / HUT | 3.7 | 2.0 | 4.5 | 3.3 | MEMLS / HUT | 3.9 | 2.4 | 3.9 | 3.2 |
| MEMLS / DMRTML | 2.4 | 1.7 | 1.2 | 1.1 | MEMLS / DMRTML | 2.0 | 1.5 | 1.2 | 1.1 |





**Table 6.** Mean bias and RMSE in brightness temperature (K) simulated by DMRT-ML (sticky and non-sticky), MEMLS (empirical and IBA-oblate) and HUT (H87, R04 and K10) forced by 189 JIM-Microstructure model combinations. Only days in the period 1st November to 31st March, where all four frequency / polarization measurements were available were included in the analysis. Bold values indicate the lowest bias / RMSE for each frequency / polarization.

| | | 2011-2012 | 19V | 19H | 37V | 37H | 2012-2013 | 19V | 19H | 37V | 37H |
|---|---|---|---|---|---|---|---|---|---|---|---|
| | DMRTML | | **0.9** | **-5.3** | 12.7 | 10.7 | DMRTML | 6.0 | **7.2** | 30.0 | 28.7 |
| Bias | MEMLS | | -7.8 | -16.3 | **-6.9** | **-6.2** | MEMLS | -9.3 | -11.4 | **-0.9** | **1.5** |
| | HUT | | -1.3 | -18.8 | 20.6 | 7.6 | HUT | **2.9** | -8.0 | 39.2 | 26.4 |
| | DMRTML | | **5.5** | **11.3** | 14.2 | 13.8 | DMRTML | 6.7 | 10.0 | 31.0 | 31.4 |
| RMSE | MEMLS | | 11.3 | 20.4 | **13.0** | 13.1 | MEMLS | 11.4 | 13.2 | **7.0** | **7.0** |
| | HUT | | 6.1 | 22.4 | 21.1 | **11.7** | HUT | **4.2** | 9.8 | 40.2 | 28.9 |



**Table 7.** Optimal microwave microstructure scale factors dependent on snow microstructure evolution function for 2011-2012 simulations, based on minimization of cost function between 1st November and 31st March in each year.

|  |  | DMRTML non | DMRTML sticky | MEMLS IBA | MEMLS EMP | HUT H87 | HUT R04 | HUT K10 |
|---|---|---|---|---|---|---|---|---|
|  | SNT | 2.0 | 0.6 | 0.5 | 0.3 | 0.9 | 0.5 | 0.7 |
| 2011-2012 | MOS | 2.3 | 1.6 | 1.7 | 1.0 | 2.6 | 1.4 | 2.2 |
|  | SNI | 4.2 | 1.3 | 1.2 | 0.8 | 1.9 | 1.1 | 1.7 |
|  | SNT | 2.3 | 0.7 | 0.7 | 0.5 | 1.2 | 1.1 | 1.1 |
| 2012-2013 | MOS | *5.0* | 1.7 | 1.6 | 1.1 | 3.2 | 2.7 | 2.9 |
|  | SNI | *5.0* | 1.5 | 1.5 | 1.1 | 2.9 | 2.3 | 2.6 |

A value of 1.0 indicates that the snow grain diameter simulated by a particular form of the snow model may be used directly in the microwave model to give the best agreement with measured brightness temperature. A value of 5.0 may not be the true minimum of the cost function as this was the largest scale factor applied.



**Table 8.** Mean bias and RMSE in brightness temperature (K) simulated by DMRT-ML (sticky and non-sticky), MEMLS (empirical and IBA-oblate) and HUT (H87, R04 and K10) forced by 189 JIM-Microstructure model combinations, with optimal microstructure scale factors from Table 7 applied. Only days in the period 1st November to 31st March, where all four frequency / polarization measurements were available were included in the analysis. Bold values indicate the lowest bias / RMSE for each frequency / polarization.

|  |  | 2011-2012 | 19V | 19H | 37V | 37H | 2012-2013 | 19V | 19H | 37V | 37H |
|---|---|---|---|---|---|---|---|---|---|---|---|
| Bias | | DMRTML | 1.1 | **-5.1** | **9.9** | 8.6 | DMRTML | 3.8 | 5.6 | **2.9** | **4.3** |
| | | MEMLS | **-0.3** | -11.3 | 13.1 | 10.5 | MEMLS | **-1.2** | **-5.1** | 6.0 | 9.1 |
| | | HUT | -1.3 | -18.8 | 19.1 | **6.2** | HUT | -4.4 | -14.3 | 18.0 | 7.5 |
| RMSE | | DMRTML | **5.5** | **11.2** | **11.4** | 11.6 | DMRTML | 4.7 | 8.9 | **5.9** | **9.2** |
| | | MEMLS | 5.7 | 15.5 | 14.0 | 12.7 | MEMLS | **3.9** | **7.6** | 9.3 | 11.7 |
| | | HUT | 6.1 | 22.4 | 19.5 | **10.7** | HUT | 6.2 | 15.4 | 19.8 | 11.2 |