# Peer review of "Microstructure representation of snow in coupled snowpack and microwave emission models"

_The Cryosphere, 2016_

## Referee Comment (RC1) · Anonymous Referee #1 · 7 Sep 2016

Report on the paper:

*Microstructure representation of snow in coupled snowpack and microwave emission models*

by Melody Sandells and co-authors

Very interesting paper on the retrieval problem of snowpack brightness temperature (Tb). The approach includes a coupling of a snow model with a microwave emission model (radiative transfer model, RTM). The authors present the results of an impressive number of combinations between snow models with 3 RTM (they used the system JIM, previously developed by R. Essery). Unfortunately, 2 well known snow models, considered as the state-of-art of snow models, Crocus and SNOWPACK, were not considered here (because not compatible with JIM). Moreover, this paper also lacks for providing some recommendations about the best (or the least worse!) of all the combinations tested!

However, this paper is publishable, but needs some clarifications for several points suggested below.

The main concern is that the reader should have a better idea of the order of magnitude of errors compared to those obtained when the RMT are driven by snow measurements (which can be considered as the reference level for RMSE and bias generally obtained in practice, under best conditions). There are a lot of papers from Finish, Canadian or American groups who provide such values. Do the scaled snow models reach the same mean level of RMSE when compared to measurements?

End of page 13 and beginning of page 14: You must clearly mention that when considering SSA measurements for retrieving the optical diameter as inputs of RTM, one must scaled the measurements too.

Also few details are given about the stratification of the snowpack considered in simulations? This is a well known significant problem for simulating Tb.

All the figures are of very poor quality, making this revision very difficult, perhaps this is an issue with building the pdf for submission, but it should be considered prior to next revision; text of scales is really too small.

*Specific comments*

I am not sure about the interest of giving all the equations in Section 2.3, especially if there are errors! See Eq. 7: (1-*f*) should be in numerator, i.e, factor of (es-eb).

Eq. 8 "it is given by the largest solution to…" largest of what?

Page 7 beginning of the Section 2.4: Why to combine HUT-MEMLS? not clear.

Table 6 : The too much detailed analysis of the differences obtained here are probably not significant?  A more synthetic analysis of results should be presented: mainly RMSE identical for the 3 models, large differences between years?

Again, page 11, line 17: "the lowest RMSE in brightness temperature is reduced, as shown in Table 8 "  probably not statistically significant?

Page 12, line 28 : "For DMRT-ML, consideration of the stickiness is imperative, with the choice of microstructure evolution model of secondary importance. " This statement is in agreement with Roy et al. (2013).

Page 13 "whereas the range in Brucker et al. (2011) was 0.25-0.4 for snow density between 100 and 400 kg m $^3$. "  In the Brucker's paper, it is the reverse. The scaling factor noted Beta = 0.63, and this sentence refers to :  A = [2/3 Beta (1 –*f*)]. Thus, when the density is 100 kg/m$^3$, A = 0.4 and when the density = 400 kg/m$^3$, A = 0.24.

Figure 2: I cannot read on the this figure if ice crusts have occurred during these two years (i.e precipitations during winter with T>0) ?  This has certainly been observed if this was the case. Please clarify.

Figure 3 : A minor observation for the SWE measurements using the GMON:  the authors observed a typical artefact of this instrument at the end of the season, when the soil is very wet due to the melting snow, leading to a strong anomaly in measurements (very high values in what is interpreted as SWE, while there is no more snow!).

Table 4 could be interesting in percentage?

---

## Referee Comment (RC2) · G. Picard (Referee) · 20 Sep 2016

It is more and more evident that the exploitation of passive microwave data to infer snow mass and other properties requires auxiliary data to overcome the general underestimation of the retrieval problem. Using microwave emission models is one promising way to provide such information. However these models need as input detailed information on the snowpack including the snow microstructure ($\sim$grain size in this paper) that are unavailable from observations. Snow evolution models suing meteorological forcing to predict time evolution of the snow physical properties are able to provide this. Because of the variety of parameterizations, modeling approach of the microstructure and existing implementation of models, an important question is which combination(s)

[Figure]

of models and parameterizations to choose in retrieval algorithms or data assimilation schemes. This paper uses an elegant method to explore the behavior and performance of coupled, snow microstructure evolution / snow microwave emission models. The originality of this paper lies in the use of a large ensemble of model-physics, following pioneer work on snow evolution only models by Essery et al. (2013), which contrast to numerous other studies that are usually limited to a few model combinations and parameterizations. This study uses data from the Arctic Sodankyla site in Finland only which is limited with respect to effort put on the size of the ensemble on the modeling side. An important conclusion of this study – valid at least for this test-site – is that improvements should be focused on snow evolution models (rather than the microwave models). Overall, the paper is well written, pleasant to read and reaches its target.

The methodology is original and certainly promising to learn more about the models and their coupling because it provides complementary information compared to the widely-used traditional "calibration/validation" approach. This is attractive, but it seems also to be limited in two ways. First the generation of the ensemble is based on subjective choices for the parameterizations, models, etc, and in some case impact the results. Some conclusions of the paper are therefore specific to these choices and may be a little bit misleading when the different models are not treated the same way (they cannot be treated equally because of their intrinsic differences for the microstructure). The difficulty for the reader is to detect the dependency to the choices because it is often hidden by the complexity of the models (and their coupling). In addition, by treating on the same level the empirical parameterizations deduced from specific sites (some of which being in Finland as the study) and the physically-based models, the methodology deprives itself of the accumulated expert knowledge on the complex physics of the snow evolution and microwave emission. Only the numerical performance on the study site is the criteria used by this methodology. For the development of a particular algorithm for operational applications, this is certainly an excellent pragmatic strategy, but the paper aims at providing general recommendations on priorities of model development. The second limitation is that although the method is very powerful to inform on

the performances of the models, it is unable (or not used here) to identify the processes responsible of the uncertainties. In other words, the main conclusion of the paper by pointing where the accuracy is insufficient is somewhat expected (though here it is demonstrated which is better than "expected") but do not provide information on how to improve these processes, which the hard part of the task. What would it mean to the modeling community if the conclusion of this statistical method was that an empirical model performs better than a physical model ?

These two limitations are mostly inherent to the methodology, not to the paper which is acceptable or could only marginally be improved on these aspects. A part of the results section (see below for details) seems to be particularly sensitive to the choices to generate the ensemble. For DMRT-ML I would recommended to remove the none-sticky case which has been shown to be inadequate in several studies, and for instance use values proposed in studies (e.g. Loewe and Picard, 2015 and Roy et al. 2013 in their discussion). For MEMLS, only two parameterizations of the scattering coefficient have been used for MEMLS while the matlab code proposed a dozen of them. Unless there is a reason (e.g. MEMLS's authors recommendation), it would be fair to explore all of them. This is only a few suggestions, any change that improve the objectivity of the choices, will improve the strength of the conclusions.

Other minor comments are added below. Then, this paper will be worth publishing because of its originality.

Detailed comments:

P1L5: "(JIM)" is needed for reference at the end of the abstract.

P2L7: "only". Photogrammetry and altimetry seem to be good (better?) candidates.

P2L9: "because of the because"

P2L30: "carried out Tedesco and Kim"

P3L20: the information in parenthesis is not clear

P5 Eq 6: add "i" in subscript to omega

P5L23: incidence-> zenith

P6 L24: 1) Is the Eq 10 only valid for average angle or this assumption is used to perform the integral ? 2) Please also check if this is the angle or the square sine/cosine of the angle that is averaged. 3) At last, I'd recommend to quote the original statement (in the reference paper) to avoid attributing this (possibly wrong) statement to your study. Or better, provide your own demonstration in Annex.

P7 L3: "Eps_eff" should be defined, here.

P7 Eq 13: "d_o" should be defined.

P8L1. It is not immediately clear where 189 comes from (3 times 63 I suppose). Maybe a slight reformulation would help.

P8L14: "there where"

P8L23: change of the state of the ground is not taken into account whereas it is an important factor in the arctic environment. Could you add information on this and provide some hints on the impact on the simulation performances ?

P9L25: this results seem dependent on the scaling factor for the precipitation. Another factor would give in different results, isn't it ? If yes, maybe better to remove this part.

P10L11 – L34 and Figure 5. The results on the spread presented in this part and in the figure depend on the choices of the parametrization and these choices seem to me unfair, i.e; orientated so that one model appears to behave a very different way from the others. This difference is however only the consequence of the choice 1) of considering the stickiness as a free parameter and 2) of the particular stickiness values. The very large value (ie. None-sticky case) is known to be unrealistic based on several recent studies. Note also that the Tsang's group usually uses values of 0.1 or 0.2. The fact that DMRT theory has two parameters to describe the microstructure

while others (apparently) use only one, does not mean that any choice of these two parameters represent snow and are valid. Conversely, why only two parameterizations of Ks in MEMLS has been used while ∼12 difference ones are available in the code ? For HUT, why the numbers appearing in equations 13, 14 and 15 (which are no more than parameters of these equations recommended by some authors) are not freely changed in this study (e.g. +-20%) to reflect the treatment in DMRT ? These choices are subjective and have too much consequences on the conveyed message of this part. This is not critical for the paper as this has no impact on the major conclusions but it suggests that the models are very different, while I thing this is mostly due to the choice. My recommendation is to narrow the range of stickiness in DMRT-ML and to explain why all the MEMLS parameterizations have not been used (or if possible/relevant to used them)

P12L12 – 15. References are needed to support the statements.

P12L25: The given reason is probably not valid, DMRT-ML predicts exactly the same propagation (within the layers) between the two polarizations (isotropic medium). This is the expected behavior for a random medium made of spheres and is indeed a sanity check used to verify the implementation. The difference in the terms of the phase function for H and V polarization in DMRT-ML are purely geometrical and the difference is canceled by the subsequent numerical integrations whereas in IBA MEMLS, it is canceled by analytical integration (or more precisely, it is removed by referring to the physical principles, the same used in DMRT-ML for the sanity check). The difference between TbH and TbV in DMRT-ML as well as in MEMLS and HUT is solely due to the interfaces. The scattering plays a role, but just because of the interaction between the volume and the interfaces, not because of difference of propagation in the volume.

A possible reason of the difference between DMRT-ML and the other models is because the initial choice of scaling (=none, with none-sticky) used in DMRT-ML makes the model really far from the observations, so that the scaling results in a general improvement.

P13L28. Please distinguish here H and V polarizations.

P14L8. Update the reference.

Figure 7. I suggest to add (light) lines between related points. In addition, the legend is difficult to understand. Please add a reference for "cluster analysis" or use a simple wording to explain what it is. Remove the external reference for the value on the x-axis and if possible add a few words to make the Figure legend more self-sufficient.

Table 3. Ratio is well not defined and difficult to understand. Why not "Comparison of grain diameter simulated by different microstructure models. The mean and max ratio between pair of models is given in columns." ? Please reformulate.

Table 5. Similarly "Ratio of mean brightness temperature ranges" is not clear. Also: would be "pair" more correct than "two" ?

---

## Author Comment (AC1) · 3 Nov 2016

Please note that the response to reviewer comments are given in the supplement

Please also note the supplement to this comment:
http://www.the-cryosphere-discuss.net/tc-2016-181/tc-2016-181-AC1-supplement.pdf

---

## Author Comment (AC2) · 3 Nov 2016

Please note the response to review comments are included in the supplement

Please also note the supplement to this comment:
http://www.the-cryosphere-discuss.net/tc-2016-181/tc-2016-181-AC2-supplement.pdf

---

## Author Response (AR1)

Response to reviewer 1:

We thank the reviewer for their supportive and insightful comments. In the first paragraph the reviewer mentions that this study does not include Crocus and SNOWPACK. These would be worthy to include in such a study, but would add the complication of differences in layering structure. Here, we have chosen an Eulerian (fixed layer) grid structure to allow better intercomparison of the internal snow physics process representation. Additionally, remote sensing retrieval applications are unlikely to be based on snowpack models of more than 5 layers, which rules out Crocus and SNOWPACK for operational, large-scale use.

The reviewer also mentions that this paper does not provide a recommendation as to the least worst model combinations. This is deliberate. The optimum choice depends on the end application (included in the discussion), but also could be totally different for an alternative site. We urge the snowpack and remote sensing community to undertake synergistic field campaigns to collect data at alternative sites with different snow conditions to allow this study to be repeated elsewhere.

With regards to the review points:

*The main concern is that the reader should have a better idea of the order of magnitude of errors compared to those obtained when the RMT are driven by snow measurements (which can be considered as the reference level for RMSE and bias generally obtained in practice, under best conditions). There are a lot of papers from Finish, Canadian or American groups who provide such values. Do the scaled snow models reach the same mean level of RMSE when compared to measurements*?

This is not a straightforward question to answer as even the snow measurements are usually scaled, and it will depend many things. The following paragraph has been added to the discussion:

Here, the lowest bias and RMSE for unscaled microstructure simulations were -6.9 to +6.9K and 4.2 to 12.2K respectively, but depended on microwave model, frequency and polarization. In an attempt to put these results into context, there are a number of studies that have quantified brightness temperature simulation errors for these models. These fall into different categories, depending on sensor characteristics, the source of the evaluation data (ground-based, airborne, satellite) and presence of ice lenses (Derksen et al. 2012), the treatment of the snow microstructure Picard et al. 2014), snow type, observation angle and the specific electromagnetic model Tedesco and Kim (2006) and the underlying substrate (Lemmetyinen et al., 2009, Derksen et al. 2014). Examples of unscaled field observations of microstructure compared with ground-based observations include the HUT simulations of Derksen et al. 2012 who found an RMSE of 10-34K and Rutter et al., 2014 who found a bias of 34-68K that was reduced to < 0.6K upon application of grain scale factors of 2.6-5.3. Scaling, or best-fit relationships were used by Durand et al., 2008 (mean absolute error 3.1K at V-pol and 9.3K at H-pol), Montpetit et al., 2013 (RMSE 8-20K), Brucker et al., 2011 (RMSE 1.5K), Picard et al., 2014 (RMSE 1-11K) and Roy et al., 2013 (RMSE 12-16K). However, in some cases the frequency-dependent results have been combined and in others kept separate.

*End of page 13 and beginning of page 14: You must clearly mention that when considering SSA measurements for retrieving the optical diameter as inputs of RTM, one must scaled the measurements too.*

This has been added to line 35: This is highlighted by the treatment of field observations of specific surface area to derive optical diameter. Even these require some form of scaling (e.g. Montpetit et al., 2013, Picard et al., 2014, Rutter et al., 2014).

*All the figures are of very poor quality, making this revision very difficult, perhaps this is an issue with building the pdf for submission, but it should be considered prior to next revision; text of scales is really too small.*

Please accept our apologies for this, we will endeavour to ensure good print quality. Figure 4 has been changed so that the two graphs are vertically aligned and larger. The font size on Figure 6 has been increased.

With regards to the specific comments:

*I am not sure about the interest of giving all the equations in Section 2.3, especially if there are errors! See Eq. 7: (1-**f**) should be in numerator, i.e, factor of (es-eb).*

We included these equations because a comparison between scattering coefficients has been missing in other publications and it highlights the empirical versus physics origin of these different models. Thank you for identifying this error, the typo has now been corrected.

*Eq. 8 "it is given by the largest solution to…" largest of what?*
Changed to 'the largest of the two solutions to the quadratic equation:'

*Page 7 beginning of the Section 2.4: Why to combine HUT-MEMLS? not clear.*
This has been changed to:

Interfacing of the various model inputs and outputs was enabled through the development of the ensemble framework, via a combination of shell script and Octave/Matlab code. The DMRT-ML model was run from the shell script, which subsequently calls an Octave / Matlab script to run HUT and MEMLS. HUT and MEMLS run alternately in this framework as the soil parameters (common between DMRT-ML and HUT) are used to calculate soil reflectivity in HUT, which is then used as the lower boundary condition in MEMLS.

*Table 6 : The too much detailed analysis of the differences obtained here are probably not significant? A more synthetic analysis of results should be presented: mainly RMSE identical for the 3 models, large differences between years?*

It is important to show the variation between models and years, and between frequencies and polarizations. Whilst the scaling may not have a large overall effect, in reality it improves the largest errors, but has a detrimental effect on the smallest

errors. It's important to highlight that scaling is not a universal solution, and we need a better understanding of the physics instead.

*Again, page 11, line 17: "the lowest RMSE in brightness temperature is reduced, as shown in Table 8 " probably not statistically significant?*
This phrase has been removed, and we focus only on individual models.

*Page 12, line 28 : "For DMRT-ML, consideration of the stickiness is imperative, with the choice of microstructure evolution model of secondary importance. " This statement is in agreement with Roy et al. (2013).*

Reviewer 2 asked for non-sticky simulations to be removed as this point can be considered a given. The Roy et al. (2013) reference is used as justification for this in section 2.3.1. Instead, stickiness values of 0.1 and 0.2 have been used.

The paper has been amended to say:
Roy et al., (2013) and Löwe and Picard (2015) showed that non-sticky representation in DMRT-ML is inappropriate. For this model ensemble, two DMRT-ML varieties have been chosen to capture the range of brightness temperatures simulated: "DMRT less sticky" (tau = 0.2) and "DMRT very sticky" (tau = 0.1). These two values represent reasonable values used by others (e.g. Tsang et al, 2007, Shih et al.,1997).

*Page 13 "whereas the range in Brucker et al. (2011) was 0.25-0.4 for snow density between 100 and 400 kg m 3. " In the Brucker's paper, it is the reverse. The scaling factor noted Beta = 0.63, and this sentence refers to : A = [2/3 Beta (1 –f)]. Thus, when the density is 100 kg/m3, A = 0.4 and when the density = 400 kg/m3, A = 0.24.*
We did not intend to imply an inverse relationship, merely that the density range of 100-400 kg m^-3 corresponds to a scaling range between 0.25 and 0.4. However, we will report the scaling range to be between 0.4-0.25 to remove ambiguity. 0.25 was reported in the paper, but 0.24 is calculated from the numbers given. Presumably limiting beta to two significant figures accounts for the difference here.

*Figure 2: I cannot read on the this figure if ice crusts have occurred during these two years (i.e precipitations during winter with T>0)? This has certainly been observed if this was the case. Please clarify.*
The following text has been added to section 2.5:

November rain events occurred in both years, and also in early December in 2011-2012. Layers with melt-freeze polycrystals and other melt forms were detected in snow pit observations during both seasons.

*Figure 3 : A minor observation for the SWE measurements using the GMON: the authors observed a typical artefact of this instrument at the end of the season, when the soil is very wet due to the melting snow, leading to a strong anomaly in measurements (very high values in what is interpreted as SWE, while there is no more snow!).*

This is an excellent point. The SWE > 0 points have been removed from Fig 3 when snow depth = 0, and the caption amended to reflect this processing to include:
Note that erroneous positive SWE observation points have been removed at the end of the season when snow depth is zero, as this is a sensor artifact related to soil moisture changes.

*Table 4 could be interesting in percentage?*

Percentages have been added to Table 4. The text has been amended to say:

The mean absolute error and mean relative difference for these simulations are presented in Table 4. SNT had the lowest bias (0.12mm) in 2011-2012, whereas SNI had the lowest bias (-0.14mm) in 2012-2013. Bulk grain diameter simulated by the microstructure models lead to a mean difference of between -53% and +45% relative to the observations.

Response to Reviewer #2: G. Picard

We thank Dr. Picard for his thorough and rigorous review of our paper. It is stated that the two main limitations of the study are inherent to the methodology and the paper is worth publishing regardless. However, we would like to respond to these comments on the limitations.

Firstly, it is true that the empirical models and models based on physics are given equal consideration. This was done to reflect the way that the models are used in the literature. Different models have different complexities due to their different intended applications. Here we intended to present this methodology as a way of identifying which processes need to be considered carefully in coupled snowpack-emission models. For processes that have little sensitivity, it doesn't matter which representation is chosen. For others with large sensitivity, it really does, and even the representations we do have are not up to scratch.

This leads on to the second limitation, that the paper does not address the physical reasons for the differences, merely which ones could be selected for further study. Investigating the physical reasons for the difference is very difficult to do within the framework of the study. It is touched upon with the snowpack component, which is computationally set up to do this thanks to Essery (2015), which brings all representations together. At present there is no equivalent electromagnetic model in the literature. This particular study is no substitute for rigorous intercomparisons of electromagnetic models such as Pan et al. (2016) and Löwe and Picard (2015). Instead, this study illustrates how future studies might be constructed. This would be substantially easier to do with a common electromagnetic model framework.

However, the potential for this study to be interpreted as 'empirical is better' is concerning. An empirical model may give a better performance at a specific site in a specific year but in no way should this be taken as a general recommendation from this paper. Nor does it help us understand why. It could be good enough for a specific application. It won't be good enough for all applications. Nor is it likely to be improved upon. A comment on this has been made in the discussion (please see later response to comment *P13L28)*

With regard to the parameter choice, we are satisfied that the literature demonstrates sufficiently that non-sticky parameterisations should never be used. We have removed the non-sticky simulations, and instead will look at the two reasonable values of stickiness applied in the literature, namely 0.1 and 0.2. This makes little difference to Figure 5, because it is dominated by the extreme stickiness = 0.1 range. It does, however, affect grain scale factor, Tb bias and RMSE, as can be expected. For the MEMLS simulations, we chose IBA as the physics-based electromagnetic model within MEMLS and the empirical scattering coefficient that is the default recommended parameterization. These are well documented and commonly used. As highlighted, there are a number of other representations of scattering coefficients (and even within the scattering coefficient model: IBA has 3 treatments of the effective permittivity and we have chosen only one). Wiesmann and Mätzler (1999) indicated that several empirical fits were available, which are versions included in the code. However, we used only the two recommended versions, commonly used in the literature, to avoid the discussion being dominated by inter-MEMLS comparisons or

disproportionately by empirical representations.

Specific comments:

*P1L5: "(JIM)" is needed for reference at the end of the abstract.*
Acronym has been replaced with Jules Investigation Model
*P2L7: "only". Photogrammetry and altimetry seem to be good (better?) candidates.*
Photogrammetry has great potential for catchment scales but isn't feasible for global scales. The text has been amended to: Microwave, altimetry or coarser-scale gravity satellite sensors offer the only feasible way to measure snow mass or depth on a global scale, with microwave observations spanning the longest time scale of these. However, microwave algorithms….
*P2L9: "because of the because"*
Removed 'because of the'
*P2L30: "carried out Tedesco and Kim"*
'by' added
*P3L20: the information in parenthesis is not clear*
Parenthesis removed, next sentence begins: As this earlier study did not incorporate snow microstructure changes, JIM was coupled with three microstructure evolution functions for this study,…
*P5 Eq 6: add "i" in subscript to omega* [change made]
*P5L23: incidence-> zenith* [change made]
*P6 L24: 1) Is the Eq 10 only valid for average angle or this assumption is used to perform the integral ? 2) Please also check if this is the angle or the square sine/cosine of the angle that is averaged. 3) At last, I'd recommend to quote the original statement (in the reference paper) to avoid attributing this (possibly wrong) statement to your study. Or better, provide your own demonstration in Annex.*
The actual calculation in MEMLS is a triple integral: an integration over the angular-dependent factor $I \sin^2 \chi$ (Mätzler and Wiesmann, 1999, eqn 9) to find the azimuth-averaged phase function, then an integration over the zenith and scattering angles according to eqn 11a of Mätzler and Wiesmann, 1999, eqn 11a. However, this is not so relevant to the discussion in light of other changes, and the section is stripped back to just the definition of the phase function without the details of calculation.
*P7 L3: "Eps_eff" should be defined, here.*
Effective permittivity was not defined for DMRT-ML, however, here a precise reference for this has been given. The exact definition will be of limited interest and more suited to a rigorous intercomparison. The text has been amended to: It should be noted that the choice of oblate grains also affects the effective permittivity in I, represented by an empirical, density-dependent effective permittivity (Wiesmann and Mätzler, 1999, equations 45-47) for this case.
*P7 Eq 13: "d_o" should be defined.*
This has been added in the second sentence in the HUT section: These are nominally suited to different grain diameter ($d_0$) ranges, with some overlap between them.

No distinction is made here regarding microwave grain diameter vs maximal grain extent. This is left to the discussion section.
*P8L1. It is not immediately clear where 189 comes from (3 times 63 I suppose). Maybe a slight reformulation would help.*
The text has been amended to: Meteorological data are used to drive the 189 configurations of JIM (3 microstructural models for each of the 63 snowpack

parameterizations).

*P8L14: "there where" [where -> were]*

*P8L23: change of the state of the ground is not taken into account whereas it is an important factor in the arctic environment. Could you add information on this and provide some hints on the impact on the simulation performances ?*

This is not considered important for this study because the permittivity measurements were so stable for the 1$^{st}$ December – 31$^{st}$ March period chosen. The evaluation was also constrained to a similar period, and the observations were very limited prior to December. The state of the ground is an interesting topic, and earlier stages of this study involved optimization of the soil permittivity, but raised many questions about the methodology. This will be revisited in the future as permittivity measurements are not commonplace but allowed us to simplify this study in this regard.

*P9L25: this results seem dependent on the scaling factor for the precipitation. Another factor would give in different results, isn't it ? If yes, maybe better to remove this part.*

The scaling factor is rather critical in terms of the snowpack modelling. It is essential to compensate for precipitation sensor undercatch as simulation of the correct temperature gradients rely on good simulation of snow depth. If these are wrong then the microstructural evolution functions (at least SNTHERM and SNICAR as these depend on the temperature gradient, whereas MOSES does not) will also be incorrect. The impact of the precipitation scaling factor itself on the microstructure is outside the scope of this study, but this modelling approach could be used to look at it.

*P10L11 – L34 and Figure 5. The results on the spread presented in this part and in the figure depend on the choices of the parametrization and these choices seem to me unfair, i.e; orientated so that one model appears to behave a very different way from the others. This difference is however only the consequence of the choice 1) of considering the stickiness as a free parameter and 2) of the particular stickiness values. The very large value (ie. None-sticky case) is known to be unrealistic based on several recent studies. Note also that the Tsang's group usually uses values of 0.1 or 0.2. The fact that DMRT theory has two parameters to describe the microstructure while others (apparently) use only one, does not mean that any choice of these two parameters represent snow and are valid. Conversely, why only two parameterizations of Ks in MEMLS has been used while _12 difference ones are available in the code ?*

*For HUT, why the numbers appearing in equations 13, 14 and 15 (which are no more than parameters of these equations recommended by some authors) are not freely changed in this study (e.g. +-20%) to reflect the treatment in DMRT ? These choices are subjective and have too much consequences on the conveyed message of this part. This is not critical for the paper as this has no impact on the major conclusions but it suggests that the models are very different, while I thing this is mostly due to the choice.*

*My recommendation is to narrow the range of stickiness in DMRT-ML and to explain why all the MEMLS parameterizations have not been used (or if possible/relevant to used them)*

The recommendation here has been followed, with the removal of non-sticky DMRT and simulation of stickiness = 0.2. In fact, the DMRT-ML simulation range is dominated almost entirely by the impact of microstructure for the stickiness = 0.1 case. The range for stickiness = 0.2 only is much narrower. The new stickiness range does, however, have an impact on the mean bias and RMSE for DMRT-ML.

*P12L12 – 15. References are needed to support the statements.*

References added, and SNICAR description changed to state that it is an albedo model.

*P12L25: The given reason is probably not valid, DMRT-ML predicts exactly the same propagation (within the layers) between the two polarizations (isotropic medium). This is the expected behavior for a random medium made of spheres and is indeed a sanity check used to verify the implementation. The difference in the terms of the phase function for H and V polarization in DMRT-ML are purely geometrical and the difference is canceled by the subsequent numerical integrations whereas in IBA MEMLS, it is canceled by analytical integration (or more precisely, it is removed by referring to the physical principles, the same used in DMRT-ML for the sanity check). The difference between TbH and TbV in DMRT-ML as well as in MEMLS and HUT is solely due to the interfaces. The scattering plays a role, but just because of the interaction between the volume and the interfaces, not because of difference of propagation in the volume.*

*A possible reason of the difference between DMRT-ML and the other models is because the initial choice of scaling (=none, with none-sticky) used in DMRT-ML makes the model really far from the observations, so that the scaling results in a general improvement.*

This comment is largely accepted. However, with a narrower range of stickiness, the DMRT-ML results are improved further. The improvements in DMRT-ML seem to be across the board, but MEMLS involves more of a trade-off. This is probably an artifact of the methodology. Without scaling, MEMLS gives reasonable results at the higher frequency, yet the scaling improves the lower frequency results at the expense of the higher frequency. In this study, the results for the different microstructure models are combined with the different versions of the same model. If the results are broken down then taking MEMLS 2011-2012 bias as an example, the optimized Tb bias improves at all frequencies and polarizations with three exceptions: IBA-MOS, where the 19H bias is less than 1K worse than the non-optimised results, and EMP-SNI combination where the improvements at the lower frequency result in poorer higher frequency simulations at both polarizations. This is sufficient to have an effect on the overall results for MEMLS. The text in this section has been changed to:

In part, this may be due to the methodology of this study as the cost function is calculated per microstructure-electromagnetic model configuration, yet the bias and RMSE are presented for each electromagnetic model family. An individual contribution can influence the group in a non-intuitive way.

*P13L28. Please distinguish here H and V polarizations.*
Looking at it in this way led to some really interesting results, with higher V-pol difference than H-pol. Table 9 has been added, which looks at all the H and V maximum differences for each unscaled microstructure – electromagnetic model combinations for year 1. Broadly, the more scattering that takes place (using grain scale factor as a proxy), the bigger the impact of the snowpack parameterizations. This is intuitive, but has been demonstrated here.

This has been added to the results:
Overall, the snowpack parameterisations with MOSES microstructure and empirical MEMLS lead to a mean difference in the 36.5GHz brightness temperature of 11K at H-pol and 18K at V-pol. The maximum difference in 36.5GHz brightness temperature

was 33K at H-pol and 54K at V-pol for the 2011-2012 season. The maximum difference between H and V polarization for all unscaled microstructure-electromagnetic model combinations is demonstrated in Table 9. Large differences in the maximum brightness temperature difference as a result of the 63 snowpack configurations occurred for the SNT microstructure. Except for DMRT-ML less sticky and HUT with MOS or SNI microstructure, the V-pol difference is greater than the H-pol difference.

This has been added to the discussion:
Although empirical MEMLS driven by MOSES was chosen as an example to demonstrate the impact of parameterisations, this was purely because of the apparent consistency between the MOSES grain diameter converted to exponential correlation length and MEMLS simulations for 2011-2012 at this site. This is not a general endorsement of empirical models, as those based on physics are expected to be more universally applicable, but the specific application of these models will dictate the balance of accuracy versus simplicity. Extending the analysis beyond this example, snow parameterizations affect other unscaled model combinations to varying degrees. Microstructure scaling factors in Table 7 can be used as a proxy for the degree of scattering in the unscaled simulations. A higher scale factor acts to increase the simulated scattering, so for a scale factor <1, too much scattering occurs in the unscaled simulations. Snowpack parameterisations have a greater impact for a higher degree of scattering, larger at V-pol than H-pol. This is because scattering is already greater at H-pol so the spread in H-pol simulations as a result of snowpack parameterisations is suppressed by the existing level of scattering. The converse applies for high scaling factors (e.g. MOS with less sticky DMRT-ML).

Both conclusions and abstract have had the equivalent of this added:
The impact of snowpack parameterisation increases as the microwave scattering increases.

*P14L8. Update the reference*. [done]
*Figure 7. I suggest to add (light) lines between related points. In addition, the legend is difficult to understand. Please add a reference for "cluster analysis" or use a simple wording to explain what it is. Remove the external reference for the value on the x-axis and if possible add a few words to make the Figure legend more self-sufficient.*
It is not helpful to add lines between related points as the black dots (1[st] February) map onto the red dots (1[st] May) for each parameterization choice (0, 1, 2). These lines would be a series of superimposed vertical lines. To remove the external reference would make this plot very hard to read as 0,1,2 for snow densification corresponds to three different densification equations. These are separate to the 0, 1, 2 options for thermal conductivity, which relate to a different set of three equations. Similarly for fresh snow density and snow hydrology. However, the caption has been changed to:

Cluster analysis of brightness temperature simulated by MOSES microstructure and empirical MEMLS for 1st February 2012 (black dots) and 1st May 2012 (red dots) according to model parameterization choices. Brightness temperature simulations are split according to the different representations for each process representation. Values 0, 1, 2 relate to parameterizations given in Essery et al., 2013 and as described in section 2.1 Where distinct clusters occur that differ between parameterizations, this

indicates sensitivity to the parameterization.

*Table 3. Ratio is well not defined and difficult to understand. Why not "Comparison of grain diameter simulated by different microstructure models. The mean and max ratio between pair of models is given in columns." ? Please reformulate.* [Done, as suggested]

*Table 5. Similarly "Ratio of mean brightness temperature ranges" is not clear. Also: would be "pair" more correct than "two" ?* [Done, similar to the above]

[revised manuscript text omitted]